# An improved Grassberger-Procaccia algorithm for analysis of climate system complexity

Chongli Di[1], Tiejun Wang[1], Xiaohua Yang[2], Siliang Li[1]

[1] Institute of Surface-Earth System Science, Tianjin University, Tianjin, 300072, P. R. China

[2] State Key Laboratory of Water Environment Simulation, School of Environment, Beijing Normal University, Beijing, 100875, P. R. China

*Correspondence to*:  Tiejun Wang (tiejun.wang@tju.edu.cn)

**Abstract.** Understanding the complexity of natural systems, such as climate systems, is critical for various
research and application purposes. A range of techniques have been developed to quantify system complexity, among which Grassberger-Procaccia (G-P) algorithm has been mostly used. However, the use of this method is still not adaptive and the choice of scaling regions relies heavily on subjective criteria. To this end, an improved G-P algorithm was proposed, which integrated the normal-based *K*-means clustering technique and Random Sample Consensus algorithm (RANSAC) for computing correlation dimensions. To test its effectiveness for
computing correlation dimensions, the proposed algorithm was compared with traditional methods using the classical Lorenz and Henon chaotic systems. The results revealed that the new method outperformed traditional algorithms in computing correlation dimensions for both chaotic systems, demonstrating the improvement made by the new method. Based on the new algorithm, the complexity of precipitation and air temperature in the Haihe River basin (HRB) in northeast China was further evaluated. The results showed that there existed
considerable regional differences in the complexity of both climatic variables across the HRB. Specifically, precipitation was shown to become progressively more complex from the mountainous area in the northwest to the plain area in the southeast; whereas, the complexity of air temperature exhibited an opposite trend with less complexity in the plain area. Overall, the spatial patterns of the complexity of precipitation and air temperature reflected the influence of the dominant climate system in the region.

## 1 Introduction

There are increasing interests in understanding system complexity, ranging from natural phenomena to social behaviors (Bras, 2015; Lin et al., 2015; Wang et al., 2016). As an open system with random external forcings and nonlinear dissipation, climate systems are highly complex (Nicolis and Nicolis, 1984; Jayawardena and Lai, 1994; Rind, 1999; Wang et al., 2015). Owing to nonlinear interactions among atmosphere, hydrosphere and
biosphere, climatic variables exhibit highly nonlinear and dynamic characteristics, which reflect the complexity of climate systems (Palmer, 1999; Rial et al., 2004; Sivakumar, 2005; Wu et al., 2010). It is thus imperative to quantitatively measure the complexity of climatic variables for understanding underlying processes. However, no common definition of system complexity exists in scientific communities, particularly from a mathematical perspective (Carbone et al., 2016). To resolve this issue, numerous concepts and methods, including chaos
theory, wavelet analysis and dynamical analysis, have been proposed to describe the complexity of climate

systems (Lorenz, 1963; Di et al., 2014; Feldhoff et al., 2014; Sivakumar, 2017; Meseguer-Ruiz et al., 2017). For instance, the chaos theory has been extensively used to characterize the chaotic and nonlinear features of climate systems (Sivakumar, 2001). Overall, previous studies based on the chaos theory revealed that the time series of air temperature and precipitation is non-stationary with abundant information. The complexity of rainfall and temperature dynamics has been widely used to indicate the extent of the complexity of climate systems (Dhanya and Kumar, 2010; Gan et al., 2014).

One of the important parameters in the chaos theory is correlation dimension, which can be used to measure the complexity and chaotic properties of variables, including precipitation and streamflow (Sivakumar et al., 2002; Dhanya and Kumar, 2011; Kyoung et al., 2011; Lana et al., 2016). Conceptually, the correlation dimension of a variable indicates the number of primary controls of the variable and thus determines the degree of freedom of the underlying process (Sivakumar and Singh, 2012). Despite the wide applications in various scientific fields, the use of the correlation dimension method is still hindered by certain limitations. For instance, the dimension method proposed by Grassberger and Procaccia (1983b) (denoted as the G-P method hereafter) is commonly used in the fields of hydrology and atmospheric science, however, its calculation procedures are still problematic (Ji et al., 2011). Specifically, the G-P method utilizes phase space reconstruction (Packard et al., 1980) and the embedding theorem (Takens, 1981) to compute correlation dimensions, which requires selection of an appropriate scaling region. The scaling region is a domain, over which an object exhibits self-similarity across a range of scales. However, the G-P method relies on visual inspections for choosing scaling regions, which is subject to human errors (Sprott and Rowlands, 2001). To tackle this problem, alternative methods have been developed to improve the original G-P method (Maragos and Sun, 1993). For example, Jothiprakash and Fathima (2013) utilized empirical equations to calculate the upper limit of scaling regions. Ji et al. (2011) applied the clustering analysis technique to determine scaling regions. However, these exsiting methods for identifying scaling regions are still not adaptive and the choice of scaling regions relies heavily on subjective criteria, and (2) the use of the least squares method for fitting straight lines to determine correlation exponents can include outliers (Cantrell, 2008) and thus is not optimal. Therefore, studies are still warranted to seek more objective and adaptive algorithms for identifying scaling regions to obtain more accurate estimates of correlation dimensions.

The primary aims of this study were two-fold. First, a new algorithm was proposed to improve the original G-P method, which integrated the methods of normal estimation, $K$-means clustering (Lloyd, 1982) and Random Sample Consensus (RANSAC; Fischler and Bolles, 1981). The classical Lorenz and Henon chaotic systems were chosen to test the effectiveness of the proposed algorithm for estimating correlation dimensions. Afterwards, the newly developed algorithm was utilized to investigate the nonlinear characteristics of precipitation and air temperature across the Haihe River basin (HRB) in northeast China. The HRB has been facing serious water shortage issues due to climate change and increasing water demand. Although previous studies have investigated climate variability (e.g., precipitation, air temperature and evaporation) in the HRB from different perspectives (Bao et al., 2012; Sang et al., 2012; Chu et al., 2010a), to our best knowledge, there are still no attempts made to quantify the nonlinear characteristics of climatic variables, especially regarding their chaotic behaviors in the HRB, which is essential for understanding the nonlinearity of the climate system in the region. Furthermore, the HRB is a diverse hydroclimatic region with many sub-watersheds of varying geographical and hydroclimatic conditions, which makes the region ideal for understanding the climate system complexity. The rest of this paper is organized as follows: Section 2 describes the calculation procedures of the

proposed algorithm, which is then tested using classical mathematical models in Section 3. Section 4 describes the data obtained from the HRB and presents the results and analysis. Conclusions are made in the last part of this paper.

## 2   Methodology

### 2.1 Algorithm for Computing Correlation Dimension

Correlation dimensions can be used to identify the complexity of dynamical systems with varying complexity degrees (e.g., low-dimensional vs. high-dimensional systems). A wealth of algorithms have been developed for computing correlation dimensions, among which the G-P algorithm has been mostly used and is also adopted in this study. The G-P algorithm uses the concept of phase space reconstruction (Packard et al., 1980) from a single-variable time series. Here, the method of delays (Takens, 1981) was employed for reconstructing phase space. Given a time series $\mathbf{X}_i$ ($i$=1, 2,…, $N$), a multi-dimensional phase space can be reconstructed as:

$$\mathbf{Y}_j = (\mathbf{X}_j, \mathbf{X}_{j+\tau}, \mathbf{X}_{j+2\tau}, \cdots, \mathbf{X}_{j+(m-1)\tau}), \tag{1}$$

where $j = 1, 2, \cdots, N - (m-1)\tau$, $m$ is the dimension of $\mathbf{Y}_j$ called embedding dimension, $\tau$ is delay time, and $\mathbf{X}_j$ is the reconstructed phase space vector.

For the $m$-dimensional reconstructed phase space, the correlation function $C(r, m)$ is defined as:

$$C(r,m) = \lim_{N \to \infty} \frac{2}{N(N-1)} \sum_{i,j=1}^{N} H(r - \|\mathbf{Y}_i - \mathbf{Y}_j\|), \quad 1 \le i \le j \le N, \tag{2}$$

where $\|\mathbf{Y}_i - \mathbf{Y}_j\|$ is the Euclidean distance between the vectors $\mathbf{Y}_i$ and $\mathbf{Y}_j$. $H(x)$ is the Heaviside function with $H(x)$=1 for $x$>0 and $H(x)$=0 for $x \le 0$, where $x = r - \|\mathbf{Y}_i - \mathbf{Y}_j\|$ and $r$ is the vector norm (i.e., radius of a sphere) centered on $\mathbf{Y}_i$ or $\mathbf{Y}_j$. Set $r_{min}$ and $r_{max}$ as the minimum and maximum distances between points, respectively (Ji et al, 2011; Lai and Lerner, 1998). If $r \le r_{min}$, none of the vector points falls within the volume element and $C(r, m)$=0. Otherwise, if $r \ge r_{max}$, all vector points fall within the volume element and $C(r, m)$=1. If there exists an attractor in the reconstructed system, $C(r, m)$ and $r$ are related through the following relationship:

$$C(r, m) \approx \alpha r^{D_2(m)}, \tag{3}$$
$$\underset{\substack{r \to 0 \\ N \to \infty}}{}$$

where $\alpha$ is a constant and $D_2(m)$ is the correlation exponent.

$D_2(m)$ is usually estimated using the least squares method by fitting a straight line through ln $r$ vs. ln $C(r, m)$. According to the relationship between $D_2(m)$ and $m$, the saturation value of $D_2(m)$ is defined as the correlation dimension. If the saturation value is low (e.g., a low correlation dimension), the system is considered to exhibit low-dimensional deterministic dynamics (i.e., a chaotic system); otherwise, the system is a stochastic one. The range, over which the straight line is fitted through ln $r$ vs. ln C($r$, $m$), is called the scaling region, where the slope is defined. Clearly, choosing an appropriate scaling region is critical for computing correlation dimensions. In previous studies, scaling regions are usually determined by visual inspections, and this will be prone to individual preferences and thus not objective. Therefore, an objective method with adaptive procedures for computing correlation dimensions is still desired.

## 2.2 Scaling Region Identification

To overcome the limitation of the original G-P algorithm for selecting scaling regions, we propose an adaptive identification algorithm of scaling regions, which utilizes the normal-based *K*-means clustering technique and the RANSAC algorithm. The use of the normal-based *K*-means clustering technique is to partition all normals of the scatter points into *K* clusters with high similarity and to remove the points that are outside of the range of the scaling region. The RANSAC algorithm was introduced to fit a straight line through the log-transformed points

obtained by the normal-based *K*-means clustering technique, which had been shown to outperform the traditional least squares method for fitting straight lines (Kyoung, 2011; Ji et al., 2011). To illustrate the advantages of using the RANSAC algorithm for linear fitting, a hypothetical example is shown in Fig. 1, which compares the fitting results obtained from the RANSAC algorithm and the traditional least squares method. The input data are sampled from a line *y*=0.5 *x*, with added noises and outliers. Here, for the RANSAC algorithm, the inliers are the

points used to fit the line; whereas, the outliers are removed from the line fitting. It can be seen from Fig. 1 that the fitting line (*y*=0.60 *x*-0.068; $R^2$=0.854) obtained from the least squares method is seriously affected by outliers and deviated from the original line *y*=0.5 *x*. By contrast, the RANSAC method is able to distinguish the inliers from outliers effectively and results in a satisfactory fitting line (*y*=0.49 *x*+0.007; $R^2$=0.990), demonstrating the advantage of using the RANSAC algorithm for linear fitting.

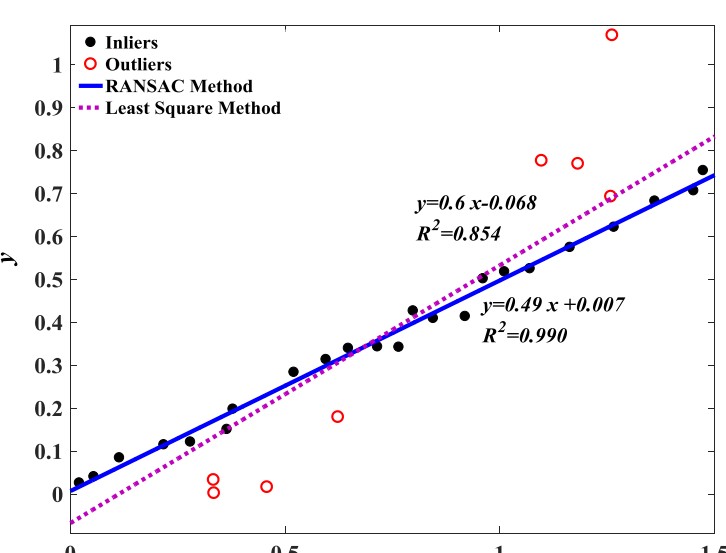

**Figure 1: Comparison of the fitted lines obtained from the RANSAC algorithm and the least squares method.**

    The flow chart of the proposed procedures for calculating correlation dimensions is given in Fig. 2, which

consists of five major steps. (1) For the time series *x*(*t*), the time delay *τ* is computed by an autocorrelation function (Liebert and Schuster, 1989). Then set the minimum embedding dimension $m_{min}$=2 and reconstruct the phase space by increasing *m* to obtain the correlation exponent function *C*(*r*, *m*); (2) The normals of the scatter points on the ln *r*~ln *C*(*r*, *m*) line are estimated via principal component analysis (Mitra et al., 2004); (3) The *K*-means clustering technique is performed on the normal set *N* with *K*=2 to obtain two different clusters. Set a

threshold value *T* to determine the angle *α* between the two clusters. If *α*>*T*, the data set with larger differences

in normals is discarded. Then, the *K*-means clustering technique is repeated on the remaining data set until *α≤T*; (4) The RANSAC algorithm is used to fit a straight line through the set of remaining scatter points; and (5) The slope of the line obtained from the RANSAC method is computed to acquire the correlation dimension $D_2(m)$ for each *m*. Finally, the saturation correlation dimension is determined using the plot $D_2(m)$ vs. *m*.

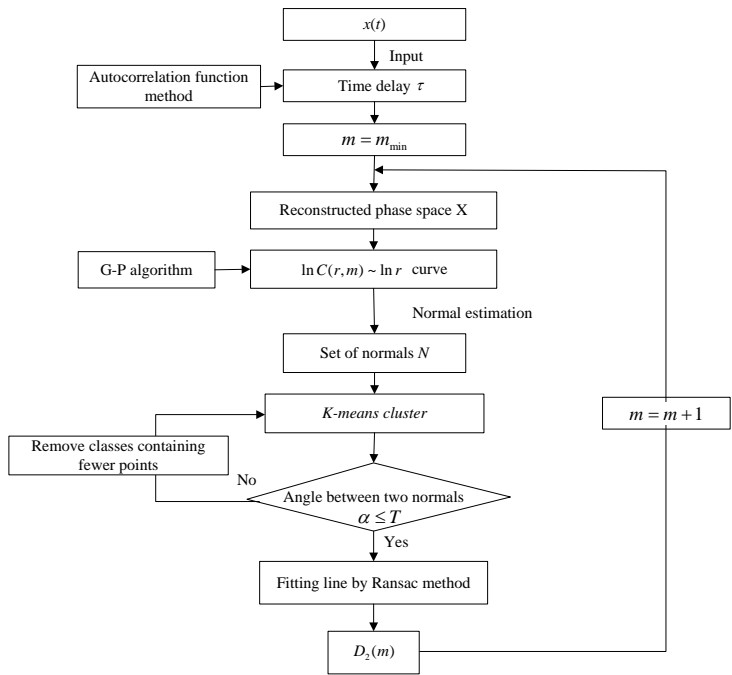

**Figure 2: Flow chart of the proposed algorithm for computing correlation dimensions (The details are listed in the text).**

**3 Verification of the Proposed Algorithm**

To test the effectiveness of the proposed algorithm, the classical chaotic models of Lorenz (1963) in Eq. (4) and

145 Henon (1976) in Eq. (5) were used. The Lorenz and Henon systems with existing theoretical correlation dimensions have been mostly studied in the past, and thus widely used to analyze the chaotic behavior in climate systems and to test the effectiveness of algorithms for computing climate system complexity (e.g., Grassberger and Procaccia, 1983a; Lai and Lerner, 1998; Ji et al., 2011).

$$\dot{x} = \sigma(-x + y),\ \dot{y} = -xz + rx - y,\ \dot{z} = xy - bz\ , \tag{4}$$

$$x_{n+1} = y_n + 1 - ax_n^2,\ y_{n+1} = bx_n\ , \tag{5}$$

where *σ*=10, *b*=28, *r*=8/3, in Eq. (4), and *a*=1.4, *b*=0.3 in Eq. (5). The theoretical dimensions of the Lorenz and the Henon systems are 2.05±0.01 and 1.25±0.02, respectively (Grassberger and Procaccia, 1983a). As a comparison, the results obtained by our proposed method were compared with the theoretical dimensions and the values obtained by another two commonly used algorithms, including the intuitive judgment method (IJM) and

155 the point-based *K*-means clustering method (PKC).

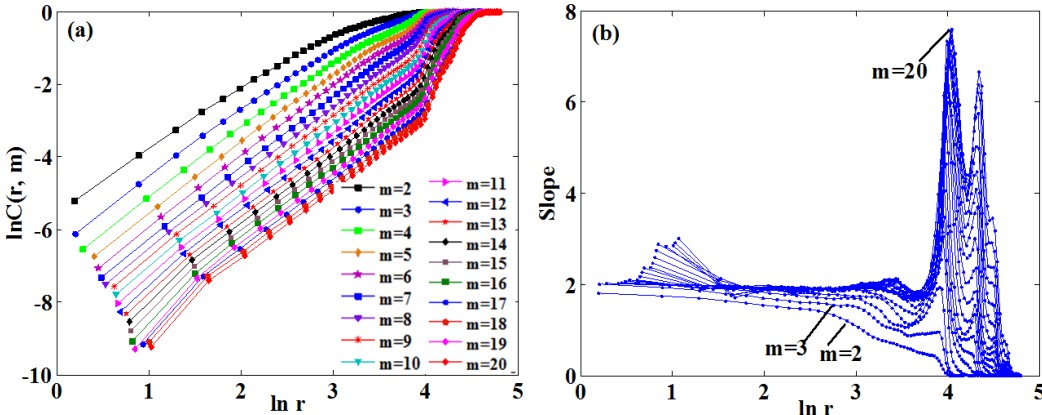

**Figure 3: Correlation integral as a function of $r$ with embedding dimension $m$ ranging from 2 to 20 for the Lorenz attractor: (a) ln $C(r, m)$ versus ln $r$, (b) the slopes of ln $C(r, m)$ versus ln $r$.**

According to the autocorrelation function, the time delay $\tau$ was determined to be 10 for the Lorenz system, with $m$ varying from 2 to 20. Figure 3a shows the relationship between ln $C(r, m)$ and ln $r$ with $m$ ranging from 2 to 20. Figure 3b shows the slopes of ln $C(r, m)$ against ln $r$ by increasing the embedding dimension $m$ (i.e., the bottom curves are associated with smaller $m$ values in Fig. 3b). The threshold value $T$ was set as 5° and $K$ was set as 2. The scaling regions of the curves in Fig. 3a were determined using the normal-based $K$-means clustering technique. As an example, an arbitrary curve was first selected from Fig. 3a, and the results are presented in Fig. 4. It can be seen from Fig. 4 that the process of the proposed method for determining the scaling region is adaptive. Specifically, for the selected curve shown in Fig. 4a, the normals of the curve were first computed based on Step 2 and the results are plotted in Fig. 4b. Different from previous $K$-means methods (e.g., the point-based $K$-means clustering method), we measured the similarity of points using the diversity between normals of different points. The reason for using the normal-based method is that the directions of normals for different points may vary considerably (see Fig. 4b); whereas, for the point-based $K$-means method, the distance between different points might be small, making it difficult to separate the points into different clusters (Fig. 3a). The obtained two separate clusters of the normals (in red and blue) are shown in Fig. 4c. If the angle $\alpha$ between the two clusters was larger than $T$, the one with larger differences in normals was discarded. Then, the $K$-means clustering technique was performed again on the remaining data set. This process was usually repeated for 2-3 times until $\alpha \leq T$ (e.g., Figs. 4c to 4e). The final scaling region was determined as shown in Fig. 4f.

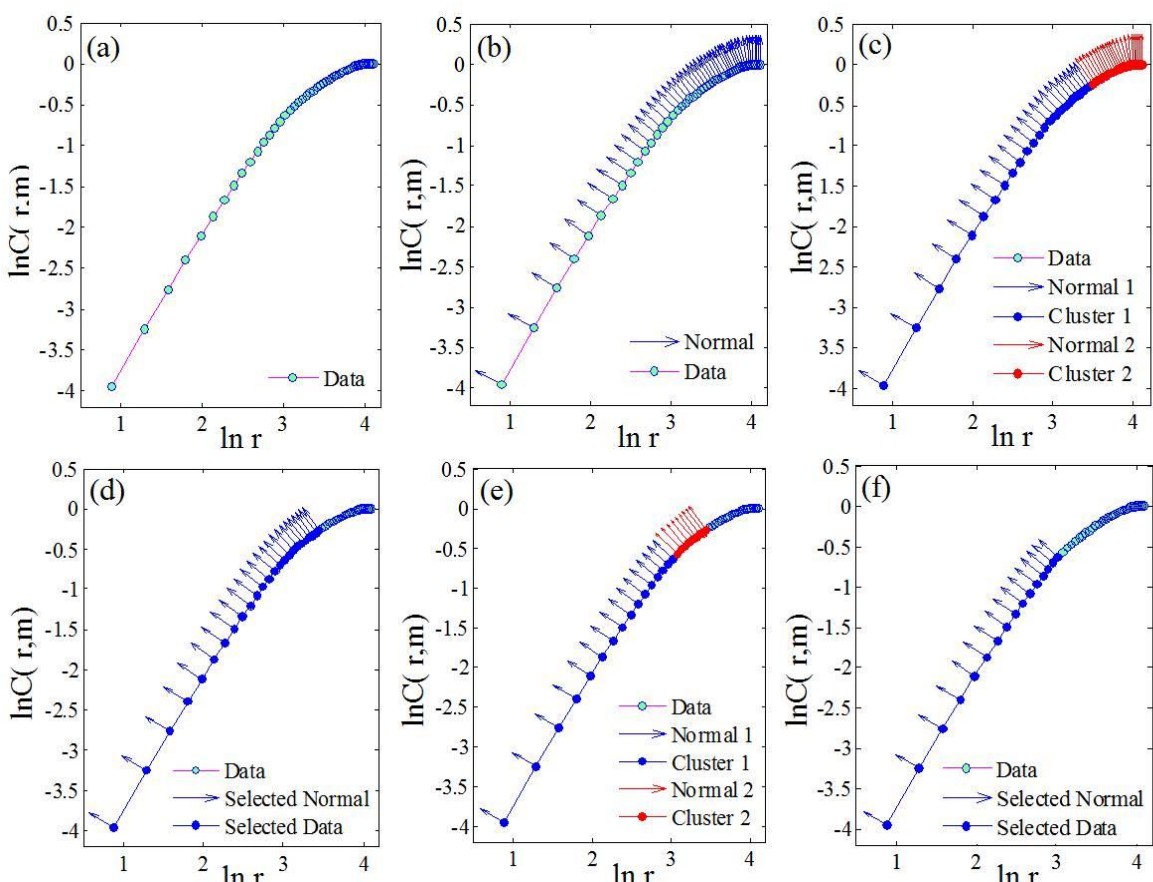

**Figure 4: Illustration of using the Normal-based K-means clustering technique for determining the scaling region. The curve shown here was randomly selected from Fig. 3.**

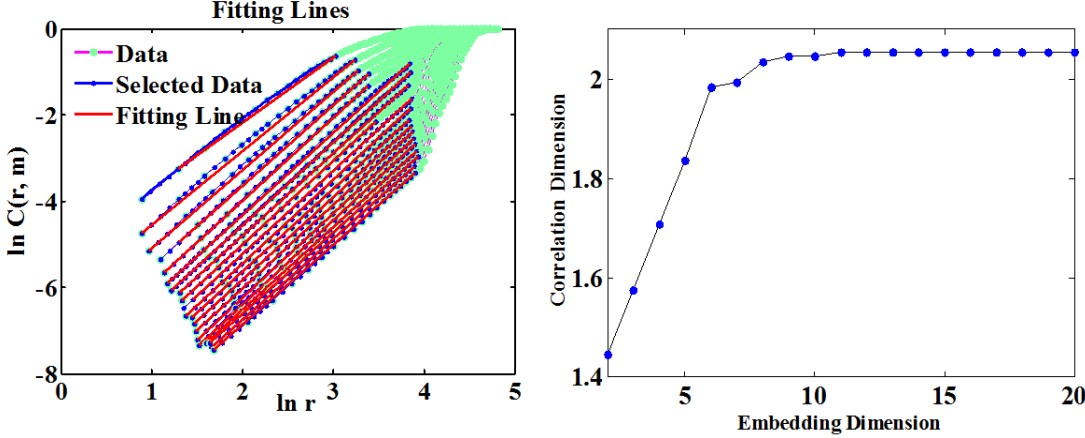

**Figure 5: The final fitted lines and the correlation dimension of the Lorenz system: (a) the final fitted lines through the scaling regions and (b) correlation dimensions as a function of embedding dimension.**

Figure 5a shows the final fitted lines through the scaling regions using the RANSAC method. The slope of the fitted line is the correlation dimension for each corresponding $m$. Figure 5b presents the graph of $D_2(m)$ against $m$ with the value of $m$ varying from 2 to 20. From Fig. 5b, we can see that $D_2(m)$ was saturated when 185 $m>5$ with the saturation value approximately equal to 2.054, which was comparable to the theoretical value of the correlation dimension for the Lorenz attractor (i.e., 2.05±0.01). Following the same procedures, the obtained

correlation dimension for the Henon attractor was 1.243, which was also close to its theoretical value (i.e., 1.25±0.02).

To verify the accuracy of our algorithm for computing correlation dimensions, the results derived from the proposed algorithm were compared with the ones obtained from the IJM and PKC methods. The IJM method was based on visual inspections to determine scaling regions (Jothiprakash and Fathima, 2013), while the PKC method integrated the *K*-means algorithm and the point-slope-error technique to determine scaling regions (Ji et al., 2011). The obtained correlation dimensions are reported in Table 1. For the Lorenz system, the differences in the correlation dimensions between the theoretical value and the ones obtained from IJM and PKC were 0.18±0.01 and 0.014±0.01, respectively; whereas, the difference was much smaller for the newly proposed algorithm (i.e. 0.004±0.01). Similar conclusions can be also made for the Henon system, demonstrating the improved performance of the proposed algorithm for determining correlation dimensions. It should be stressed that despite the improvement made by our proposed algorithm, further studies are still needed to address the issues on the computation of correlation dimensions. For example, estimation of correlation dimensions is partly dependent on the proper selection of time delay and embedding dimension; therefore, the impacts of their uncertainties should be further assessed.

**Table 1.** Comparison of the correlation dimensions derived from different methods.

| Attractor | TCD | IJM | PKC | NPA |
|-----------|-----|-----|-----|-----|
| Lorenz | 2.05±0.01 | 2.23±0.02 | 2.064 | 2.054 |
| Henon | 1.25±0.02 | 1.354±0.02 | 1.240 | 1.243 |

Note-TCD: Theoretical correlation dimension; IJM: Intuitive judgment method; PKC: Point-based *K*-means clustering; NPA: Newly proposed algorithm.

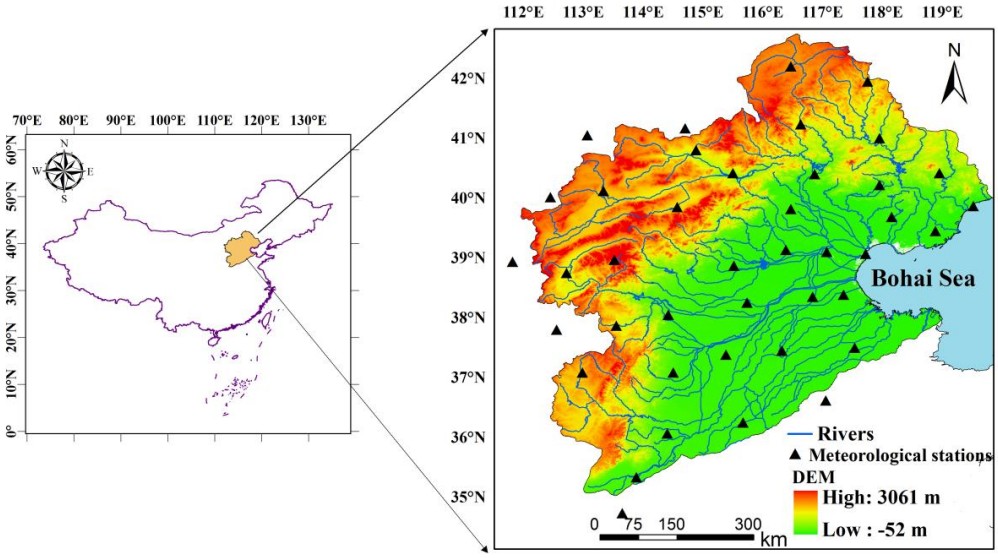

**Figure 6: Locations of meteorological stations in the Haihe River basin.**

## 4 Application, Results and Analysis

The correlation dimension method is an important diagnostic tool for understanding the complexity of natural systems with chaotic characteristics. In this section, a case study is presented to illustrate the use of the newly

developed algorithm for studying the complexity of climate systems. Specifically, the algorithm was first utilized to compute the correlation dimensions of precipitation and air temperature using time series obtained from the HRB. Afterwards, the regional patterns of correlation dimensions for precipitation and air temperature in the HRB were analyzed.

### 4.1 Study area and Data

The HRB is located in northeast China (112 °–120 °E, 35 °–43 °N; Fig. 6), which hosts one of the most important economic zones in China (White et al., 2015). Under the influences of climate change and human activities, complex water issues have become increasingly prominent in the HRB (Liu and Xia, 2004). Topography varies considerably across the area, with 22% of the total area for mountains in the western and northern parts, 40% for plains in the eastern and southern parts, and 38% for hilly areas in the central part. The regional climate in the

HRB is of a semiarid or subhumid type, with mean annual precipitation of 539.0 mm/year and mean annual temperature of 10.2°C. Mean annual precipitation increases from the mountainous areas in the west to the plains in the east, while mean annual temperature decreases along the direction from south to north. In addition, precipitation in the HRB exhibits significant interdecadal and interannual variations. To apply the proposed algorithm for computing correlation dimensions, monthly precipitation and air temperature data spanning from

1951 to 2016 were retrieved from 40 meteorological stations in the HRB and nearby areas (Fig. 6), which were operated by the China Meteorological Administration (http://data.cma.cn/site/index.html).

### 4.2 Results and Analysis

The correlation dimensions of precipitation and air temperature at all 40 meteorological stations were computed using the algorithm proposed in this study. Figure 7 shows the relationships between correlation dimension and

embedding dimension for precipitation and air temperature at five representative stations across the HRB (i.e., Beijing, Fengning, Shijiazhuang, Xinxiang and Zhangbei). The embedding dimensions of precipitation and air temperature for the five stations varied between 10 and 12. It is evident that the relationship between correlation dimension and embedding dimension for precipitation and air temperature differed among the selected stations. In genearal, correlation dimensions for precipitation showed gradual saturation processes with respective

saturation values of 2.378, 2.407, 3.055 and 2.550 for Beijing, Fengning, Shijiazhuang and Zhangbei stations (Fig. 7a), indicating chaotic dynamical characteristics of precipitation. By comparison, the correlation dimension for precipitation at the Xinxiang station increased with increasing embedding dimensions, suggesting random characteristics of precipitation. For air temperature, the correlation dimensions at the five stations also showed gradual saturation processes (Fig. 7b), suggesting low dimensional chaotic characteristics for air temperature.

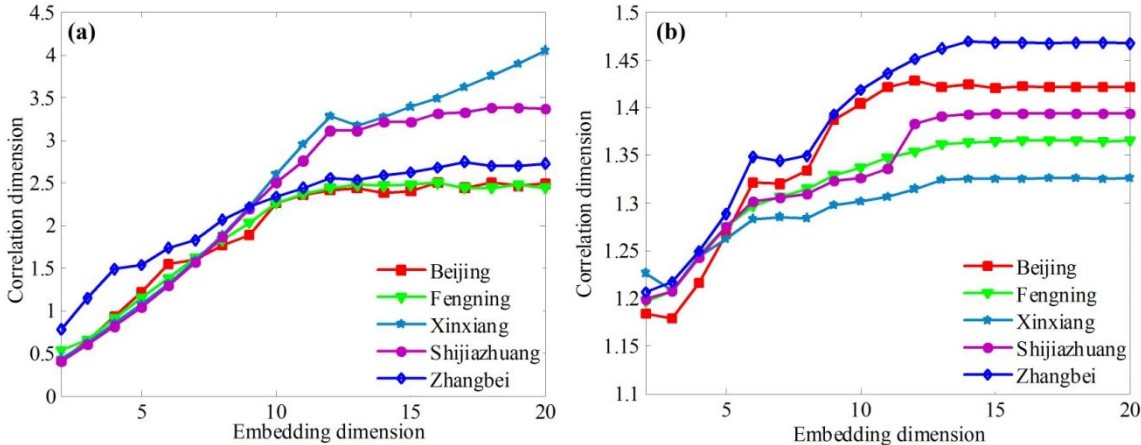

**Figure 7: Variation of correlation dimension versus embedding dimension of climate variables: (a) precipitation and (b) air temperature.**

Figure 8 presents the spatial distributions of the saturated correlation dimensions at the 40 meteorological stations for precipitation and air temperature in the HRB. For both precipitation and air temperature, the correlation dimensions varied markedly across the area. The correlation dimension for precipitation ranged from less than 3 to more than 6, while the correlation dimension was much lower for temperature (i.e., less than 2). Overall, the ranges of the correlation dimensions for precipitation and air temperature were comparable to previously reported values in other regions with similar climatic conditions (Kyoung et al., 2011; Sivakumar and Singh, 2012; Sivakumar et al., 2014). More importantly, the considerable spatial variations in the dimensionality for both climatic variables suggest the regional differences in the complexity of the climate system in the HRB. Specifically, the correlation dimension for precipitation tended to be smaller in the northwestern mountainous area with the values less than 2.5. In the central area, the correlation dimension for precipitation became larger with the values greater than 3, while precipitation in the southeastern plain area showed very high correlation dimensions with the values larger than 6. Given that correlation dimensions indicate the number of controls on the underlying process (Sivakumar and Singh, 2012), Fig. 8a suggests that precipitation processes become progressively more complex from the mountainous area to the plain area in the HRB. Interestingly, the regional pattern of the correlation dimension for air temperature showed an opposite trend with smaller values mainly located in the northern HRB, indicating more complex temporal dynamics of air temperature in the area.

The spatial pattern of the correlation dimension for precipitation in the HRB may be largely attributed to the regional flow pathway of moisture flux, which is mainly controlled by the East Asian Summer Monsoon (EASM). The HRB is located in a monsoon-dominated region, where the EASM plays a leading role in the regional meteorological system. Chen et al. (2013) showed that EASM had significant impacts on the spatiotemporal distribution of precipitation in East China. Li et al. (2017) further suggested that there was a significant correlation between precipitation and the EASM index in the HRB. Wang et al. (2011) revealed that large-scale atmospheric circulations had close relationships with precipitation patterns in the HRB by analyzing the moisture flux derived from NCAR/NCEP reanalysis data. Influenced by the large-scale atmospheric circulation, precipitation in the middle and southeast parts of the HRB is more sensitive to climate variability due to their locations closer to the ocean. This leads to the decreasing trend of precipitation from southeast to

northwest in the HRB, suggesting that the supply of moisture for precipitation in the region mainly comes from the ocean.

Partly owing to the closer geographical proximity to the ocean (Fig. 8), the EASM has a stronger impact on precipitation in the southern and central areas than in the northern part of the HRB. Furthermore, at the north corner of the HRB, the westerlies primarily affect the hydrometeorological system and thus weaken the impact of the EASM on precipitation (Li et al., 2017). In addition, other factors (e.g., topography, vegetation distribution, and human activity) may also have impacts on regional patterns of climate variables. In particular, the Yan-Taihang mountain located in the northwest HRB obstructs the vapor transport driven by the EASM, resulting in lower spatiotemporal variability in precipitation in the north part of the HRB. As a result, precipitation had higher degrees of complexity in the southern HRB, while its complexity was lower in the mountainous area in the northwest HRB. As to air temperature, the orographic effect in the mountainous area on air temperature might be stronger (Chu et al., 2010b), resulting in the higher complexity of temperature in this area. However, it should be noted that the range of the correlation dimension for air temperature from 1.0 to 2.0 suggests that two primary controls on temperature exist at all stations across the region.

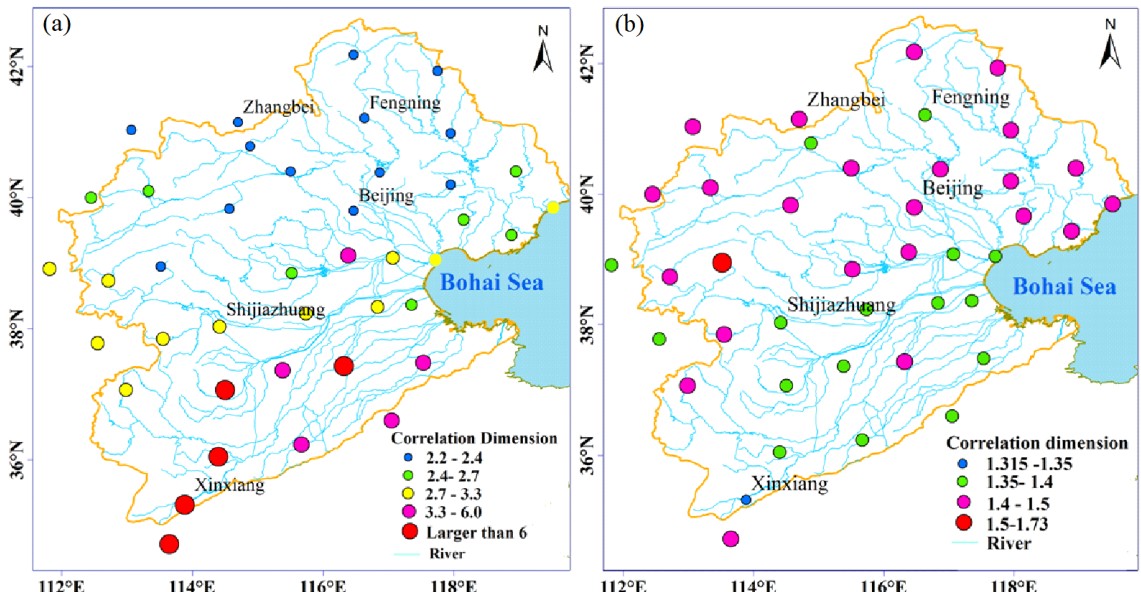

**Figure 8: The spatial distribution of the correlation dimension values for all the 40 stations: (a) precipitation and (b) temperature.**

**5 Conclusions**

In this study, the original G-P algorithm for calculating correlation dimensions was modified by incorporating the normal-based *K*-means clustering technique and the RANSAC algorithm. Using the proposed method, the spatial patterns of the complexity of precipitation and air temperature in the HRB were analyzed. The following conclusions were reached:

(1) The effectiveness of the proposed method for calculating correlation dimensions was illustrated using the classical Lorenz and Henon chaotic systems. The results showed that the new method outperformed the traditional intuitive judgment and point-based *K*-means clustering method for computing correlation dimensions.

(2) Except for few stations in the northern region, precipitation at most of the meteorological stations in the HRB showed chaotic behaviors. Specifically, the correlation dimension for precipitation showed an increasing trend from the mountainous region in the northwest to the plain area in the southeast, indicating that precipitation processes became progressively more complex from the mountainous area to the plain area. The spatial pattern of the complexity of precipitation reflected the influence of the dominant climate system in the region. Meanwhile, air temperature at all meteorological stations showed chaotic characteristics. In contrast to precipitation, the complexity of air temperature exhibited an opposite trend with less complexity in the plain area.

The modified G-P algorithm proposed in this study can be used more objectively to characterize the complexity of climate systems (and other hydrological systems, such as streamflow, soil moisture, and groundwater), and thus provide a more reliable estimate of the number of dominant factors governing climate systems. Theoretically, it can provide valuable information for optimizing the number of parameters in climate models to reduce computational demands and model parameter uncertainties. Furthermore, the findings of this study can be used for the regionalization of hydrometeorological systems in the HRB, which has important significance in prediction in ungaged areas (Lebecherel et al., 2016). It should be noted that more studies are still required to verify the present results using other nonlinear techniques, such as the Lyapunov exponent (Wolf et al., 1985) and the approximate entropy (Pincus, 1995), which might provide additional insights into climate complexity analysis.

**Data Availability**

The datasets used in this study are publicly available. The monthly precipitation and temperature data can be downloaded from the China Meteorological Administration Network (http://data.cma.cn/site/index.html). The code for computing correlation dimension can be acquired from the first author C. Di.

**Competing interests**

The authors declare that they have no conflict of interest.

**Acknowledgements**

The work was supported by the National Key R&D Program of China (No. 2016YFA0601002, No.2016YFC0401305, and No. 2017YFC0506603), and by the National Natural Scientific Foundation of China (No. 51679007 and No. U1612441). The authors would also like to acknowledge the financial support from the Tianjin University and T. Wang also acknowledges the financial support from the Thousand Talent Program for Young Outstanding Scientists for this study.

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
