# Peer review of "An improved Grassberger-Procaccia algorithm for analysis of climate system complexity"

_Hydrology and Earth System Sciences, 2017_

## Referee Comment (RC1) · Anonymous Referee #1 · 12 Oct 2017

1. The manuscript presents an improved Grassberger-Procaccia algorithm for analysis of climate system complexity, which is interesting. The subject addressed is within the scope of the journal. 2. However, the manuscript, in its present form, contains several weaknesses. Appropriate revisions to the following points should be undertaken in order to justify recommendation for publication. 3. For readers to quickly catch your contribution, it would be better to highlight major difficulties and challenges, and your original achievements to overcome them, in a clearer way in abstract and introduction. 4. It is shown in the reference list that the authors have several publications in this field. This raises some concerns regarding the potential overlap with their previous works. The authors should explicitly state the novel contribution of this work, the similarities and the differences of this work with their previous publications. 5. It is mentioned in p.1

that an improved Grassberger-Procaccia algorithm is adopted for analysis of climate system complexity. What are the other feasible alternatives? What are the advantages of adopting this particular algorithm over others in this case? How will this affect the results? More details should be furnished. 6. It is mentioned in p.2 that Lorenz and Henon chaotic systems are adopted to test the effectiveness of the proposed algorithm for estimating correlation dimensions. What are the other feasible alternatives? What are the advantages of adopting these particular systems over others in this case? How will this affect the results? More details should be furnished. 7. It is mentioned in p.2 that the Haihe River Basin is adopted as the case study. What are other feasible alternatives? What are the advantages of adopting this particular case study over others in this case? How will this affect the results? The authors should provide more details on this. 8. It is mentioned in p.3 that the normal-based K-means clustering technique is adopted to partition all normals of the scatter points into K clusters with high similarity. What are other feasible alternatives? What are the advantages of adopting this particular technique over others in this case? How will this affect the results? The authors should provide more details on this. 9. It is mentioned in p.4 that the Random Sample Consensus algorithm is adopted to fit a straight line through the log-transformed points. What are other feasible alternatives? What are the advantages of adopting this particular technique over others in this case? How will this affect the results? The authors should provide more details on this. 10. It is mentioned in p.6 that the intuitive judgment method and the point-based K-means clustering method are adopted to compare the results obtained by the proposed method. What are the other feasible alternatives? What are the advantages of adopting these particular methods over others in this case? How will this affect the results? More details should be furnished. 11. It is mentioned in p.6 that the normal-based K-means clustering technique is adopted to determine the scaling regions of the curves in Fig. 3a. What are other feasible alternatives? What are the advantages of adopting this particular technique over others in this case? How will this affect the results? The authors should provide more details on this. 12. Some key parameters are not mentioned. The rationale on the choice

of the particular set of parameters should be explained with more details. Have the authors experimented with other sets of values? What are the sensitivities of these parameters on the results? 13. Some assumptions are stated in various sections. Justifications should be provided on these assumptions. Evaluation on how they will affect the results should be made. 14. The discussion section in the present form is relatively weak and should be strengthened with more details and justifications. 15. Moreover, the manuscript could be substantially improved by relying and citing more on recent literatures about real-life case studies of contemporary soft computing techniques in hydrological engineering such as the followings: ïĄň Gholami, V., et al., "Modeling of groundwater level fluctuations using dendrochronology in alluvial aquifers", Journal of Hydrology 529 (3): 1060-1069 2015. ïĄň Taormina, R., et al., ""Neural network river forecasting through baseflow separation and binary-coded swarm optimization", Journal of Hydrology 529 (3): 1788-1797 2015. ïĄň Wu, C.L., et al., "Prediction of rainfall time series using modular artificial neural networks coupled with data-preprocessing techniques", Journal of Hydrology 389 (1-2): 146-167 2010. ïĄň Wang, W.C., et al., "Improving forecasting accuracy of annual runoff time series using ARIMA based on EEMD decomposition," Water Resources Management 29 (8): 2655-2675 2015. ïĄň Chen, X.Y., et al., "A comparative study of population-based optimization algorithms for downstream river flow forecasting by a hybrid neural network model," Engineering Applications of Artificial Intelligence 46 (A): 258-268 2015. ïĄň Chau, K.W., et al., "A Hybrid Model Coupled with Singular Spectrum Analysis for Daily Rainfall Prediction," Journal of Hydroinformatics 12 (4): 458-473 2010. 16. In the conclusion section, the limitations of this study, suggested improvements of this work and future directions should be highlighted.

---

## Author Comment (AC1) · 13 Nov 2017

We would like to thank the reviewer for his/her time and effort in reviewing our manuscript, titled 'An improved Grassberger-Procaccia algorithm for analysis of climate system complexity' (ID: hess-2017-445). Your comments and suggestions are much appreciated, which are useful to further improve our manuscript. Please see our responses in the following section.

Comment 1. For readers to quickly catch your contribution, it would be better to highlight major difficulties and challenges, and your original achievements to overcome them, in a clearer way in abstract and introduction.

Response: Thank you for the comment here. In Introduction, we have stated some of the major problems associated with the current methods for computing correlation dimensions (e.g., in lines 11-12 'the use of this method is still not adaptive and relies heavily on subjective criteria', and in lines 53-54, 'However, the G-P method relies on visual inspections for choosing scaling regions, which is subject to human errors (Sprott and Rowlands, 2001). To deal with this specific problem, we tried to find 'more objective and adaptive algorithms for identifying scaling regions to obtain more accurate estimates of correlation dimensions' (lines 59-60). Nonetheless, based on the reviewer's comment here, we will further highlight our contribution for computing correlation dimensions in Abstract and Introduction.

Comment 2. It is shown in the reference list that the authors have several publications in this field. This raises some concerns regarding the potential overlap with their previous works. The authors should explicitly state the novel contribution of this work, the similarities and the differences of this work with their previous publications.

Response: Thank you for the comment. First, the novelty of our current work as compared to previous studies is discussed in details in Abstract and Methodology. Secondly, the studies cited in the reference list (presumably with the last name of Wang) are done by others and not published by the current authors.

Comment 3. It is mentioned in p.1 that an improved Grassberger-Procaccia algorithm is adopted for analysis of climate system complexity. What are the other feasible alternatives? What are the advantages of adopting this particular algorithm over others in this case? How will this affect the results? More details should be furnished.

Response: Thank you for the suggestion. In fact, we have given some alternative methods for studying climate system complexity, such as chaos theory, wavelet analysis, and dynamical analysis (line 35). In particular, for computing correlation dimensions, we also compared our newly proposed algorithm to two other commonly used algorithms, namely the intuitive judgment and the point-based K-means clustering methods, based on two classical chaotic systems. Nevertheless, based on the reviewer's comment here, we will add few sentences to illustrate the differences among existing methods.

Moreover, to address this comment as well as following comments made by the reviewer, we feel that in a single technical paper with limited space, it is not feasible and appropriate to include every aspect existing in the field of complexity analysis,

which would deviate from the central theme of this study and make the manuscript unnecessarily excessive. In fact, there are several excellent books that are devoted to entirely discussing relevant problems, which we would like to refer the reviewer to (e.g., Bellie Sivakumar, 2017; Jayawardena, 2014). In addition, we will add those books to the reference list for the convenience of readers.

Sivakumar B. Chaos in Hydrology. Springer Netherlands, 2017.
Jayawardena   A. W. Environmental and Hydrological Systems Modelling. Taylor and Francis Group, CRC Press, 2014.

Comment 4. It is mentioned in p.2 that Lorenz and Henon chaotic systems are adopted to test the effectiveness of the proposed algorithm for estimating correlation dimensions. What are the other feasible alternatives? What are the advantages of adopting these particular systems over others in this case? How will this affect the results? More details should be furnished.
Response: Thank you for this comment. Indeed, there are other chaotic systems (e.g., the Chen system (Chen and Ueta, 1999), and the Rössler system (Rössler, 1979)). Among those chaotic systems, the Lorenz and Henon systems have been mostly used to analyze the chaotic behavior in climate systems and to test the effectiveness of algorithms for computing climate system complexity (e.g., Grassberger and Procaccia, 1983; Lai and Lerner, 1998; Ji et al., 2011), moreover, the correlation dimensions of those two systems are mostly studied. In our opinion, for the purpose of brevity and comparison among different studies and methods for computing climate system complexity, it is justified that standard systems, such as the Lorenz and Henon systems, should be adopted. Finally, the discussion on different chaotic systems is beyond the scope of this study. It would be unrealistic for us to compare all chaotic systems in a paper. Certainly, we can add more details and the following references in our revised manuscript.
References:
Chen, G., Ueta, T.: Yet another chaotic attractor, International Journal of Bifurcation and Chaos, 9, 1465-1466, 1999.
 Rössler, O. E.: An equation for hyperchaos. Physics Letters, 71A (2, 3): 155-157, 1979.
Lai Y. C., Lerner D.: Effective scaling regime for computing the correlation dimension from chaotic time series. Physica D, 115: 1-18, 1998.
Ji, C.C., Zhu, H. and Jiang, W.: A novel method to identify the scaling region for chaotic time series correlation dimension calculation, Chinese Sci. Bull., 56, 925-932, doi: 10.1007/s11434-010-4180-6, 2011.

Comment 5. It is mentioned in p.2 that the Haihe River Basin is adopted as the case study. What are other feasible alternatives? What are the advantages of adopting this particular case study over others in this case? How will this affect the results? The authors should provide more details on this.
Response: Thank you for the comment here. The reasons that we took the Haihe River

Basin (HRB) as a case study are as follows: (1) The HRB has been facing serious water shortages due to climate change and increasing water demand. Although previous studies have investigated the climate variability (e.g., rainfall, air temperature, and evaporation) in the HRB from different perspectives, to our best knowledge, there are still no attempts made to quantify nonlinear characteristics of climatic variables, especially regarding their chaotic behaviors in the HRB, which is essential for understanding the nonlinearity of the climate system in the region; and (2) The HRB is a diverse hydroclimatic region with many sub-watersheds of varying geographical and climatic conditions, which make the region ideal for understanding the climate system complexity. Certainly, we can add more details about the advantages of adopting this particular case in our revised manuscript.

Comment 6. It is mentioned in p.3 that the normal-based K-means clustering technique is adopted to partition all normals of the scatter points into K clusters with high similarity. What are other feasible alternatives? What are the advantages of adopting this particular technique over others in this case? How will this affect the results? The authors should provide more details on this.

Response: Thank you for the suggestion. The K-means clustering method has been widely used for cluster analysis, which aims to partition $n$ observations into $K$ clusters. For each cluster, each observation belongs to the cluster with the nearest mean. In this paper, in order to find a precise scaling region, we used the K-means clustering algorithm to remove the points, which were obviously located outside the real scaling region. Different from previous K-means methods (e.g., the point-based K-means clustering method), we measured the similarity of points using the diversity between points' normal, that is, the normal-based K-means clustering technique. This is because the normal directions of different points in Figure 4(a) are greatly different. By comparison, the distance between points is much less, due to the use of the logarithmic scale that makes the points more densely distributed as $\ln r$ goes backward (see Fig 3(a)). Therefore, we proposed to use the normal-based K-means clustering algorithm. As a comparison, taking the classical chaotic models of Lorenz and Henon as two examples, the results obtained by our proposed normal-based k-means method performed better than the point-based k-means method (see Table 1). To illustrate this, we will add some sentences to show the advantages of normal-based K-means method in the methodology section.

Comment 7. It is mentioned in p.4 that the Random Sample Consensus algorithm is adopted to fit a straight line through the log-transformed points. What are other feasible alternatives? What are the advantages of adopting this particular technique over others in this case? How will this affect the results? The authors should provide more details on this.

Response: Thank you for the comment. We have given the reasons for choosing the Random Sample Consensus algorithm (RANSAC) in section 2.2. As shown in section 2.2, the RANSAC algorithm outperformed the commonly used least squares method for linear fitting, based on a hypothetical example (Fig. 1).

Comment 8. It is mentioned in p.6 that the intuitive judgment method and the point-based K-means clustering method are adopted to compare the results obtained by the proposed method. What are the other feasible alternatives? What are the advantages of adopting these particular methods over others in this case? How will this affect the results? More details should be furnished.

Response: Thank you for the comment. The intuitive judgment method and the point-based K-means clustering method are two commonly used methods for identifying scaling region (e.g., Sprott and Rowlands, 2001; Ji et al., 2011). Although more comparisons can be done, additional comparisons may seem redundant. In addition, it is unrealistic to list all the comparisons in one paper.

Comment 9. It is mentioned in p.6 that the normal-based K-means clustering technique is adopted to determine the scaling regions of the curves in Fig. 3a. What are other feasible alternatives? What are the advantages of adopting this particular technique over others in this case? How will this affect the results? The authors should provide more details on this.

Response: This comment is the same as the comment 8.

Comment 10-11. 10. Some key parameters are not mentioned. The rationale on the choice of the particular set of parameters should be explained with more details. Have the authors experimented with other sets of values? What are the sensitivities of these parameters on the results? 11. Some assumptions are stated in various sections. Justifications should be provided on these assumptions. Evaluation on how they will affect the results should be made.

Response: Thank you for this comment. We rechecked the paper and found that the ranges of $r$ were missing. We will add more details in line 92 (i.e. 'Set $r_{min}$ and $r_{max}$ as the minimum and maximum distances between points, respectively (Ji et al, 2011; Lai and Lerner, 1998). If $r \leqslant r_{min}$, none of the vector points will fall within the volume element and $C(r, m)=0$. Otherwise, if $r \geqslant r_{max}$, all vector points will fall within the volume element and $C(r, m)=1$.'). Other parameters have been given in the paper. We must point out that some of the parameters in this study were determined by routinely used methods. For example, the time delay (see line 86) was determined by the autocorrelation function. Some other parameters (for example, T=5°) were determined by testing the data. In terms of the assumption about the value $r$, we will add it in our paper.

Comment 12. The discussion section in the present form is relatively weak and should be strengthened with more details and justifications.

Response: Considering that this is a technical paper, we limited our discussions for the purpose of brevity. However, if needed, we can give more details and justifications in our revised paper.

Comment 13. The manuscript could be substantially improved by relying and citing

more on recent literatures about real-life case studies of contemporary soft computing techniques in hydrological engineering such as the followings: ïA¸nˇ Gholami, V., et al., "Modeling of groundwater level fluctuations using dendrochronology in alluvial aquifers", Journal of Hydrology 529 (3): 1060-1069 2015. ïA¸nˇ Taormina, R., et al., ""Neural network river forecasting through baseflow separation and binary-coded swarm optimization", Journal of Hydrology 529 (3): 1788-1797 2015. ïA¸nˇ Wu, C.L., et al., "Prediction of rainfall time series using modular artificial neural networks coupled with data-preprocessing techniques", Journal of Hydrology 389 (1-2): 146-167 2010. ïA¸nˇ Wang, W.C., et al., "Improving forecasting accuracy of annual runoff time series using ARIMA based on EEMD decomposition," Water Resources Management 29 (8): 2655-2675 2015. ïA¸nˇ Chen, X.Y., et al., "A comparative study of population-based optimization algorithms for downstream river flow forecasting by a hybrid neural network model," Engineering Applications of Artificial Intelligence 46 (A): 258-268 2015. ïA¸nˇ Chau, K.W., et al., "A Hybrid Model Coupled with Singular Spectrum Analysis for Daily Rainfall Prediction," Journal of Hydroinformatics 12 (4): 458-473 2010.

Response: Thank you for providing the relevant references for further modification of our paper, and we have read them and we will also cite some of them in the revised paper.

Comment 14. In the conclusion section, the limitations of this study, suggested improvements of this work and future directions should be highlighted.

Response: Thank you for the comment. We will discuss the limitations of this study and suggest possible future improvements for computing correlation dimensions.

---

## Referee Comment (RC2) · Anonymous Referee #2 · 1 Sep 2018

An improved Grassberger-Procaccia (G-P) algorithm was proposed, which integrated the normal-based K-means clustering technique and Random Sample Consensus algorithm (RANSAC) for computing correlation dimensions. It is tested for computing correlation dimensions for the analysis of climate system complexity, and compared with traditional methods using the classical Lorenz and Henon chaotic systems. Proposed algorithm found to be better than the existing algorithms.

Comments: (1) Section 2.1 Algorithm for Computing Correlation Dimension may be reduced as correlation dimension is relatively old.

(2) Line 117-119 and Figure 1: Authors compared equations in terms of y =0.5x. What is R square value for both the equations and this also can be taken into consideration while judging superiority of methods.

[Figure]

(3) Line 125-140: Detailed information is provided and flow chart is also presented. Detailed information can be reduced as the flow chart is self explanatory.

(4) Figure 8: More discussion will help to understand the figure effectively.

(5) Utility of estimation of correlation dimensions for the future work in HRB can be briefly mentioned.

---

## Editor Comment (EC1) · D. Solomatine (Editor) · 6 Sep 2018

Dear authors, The "discussion" phase (HESS-D) is now finished. Thank you for providing comprehensive replies to the referees' comments and suggestions for revisions (which were very useful in my opinion). Based on the discussion, it can be concluded that the paper is at the level to be considered for HESS publication. You will now have the time and opportunity to update the manuscript accordingly, and submit the new version to the editorial office. After that it will be sent to the referees for their evaluation before publication in HESS. Good luck.

---

## Author Comment (AC2) · 6 Sep 2018

We would like to thank the Referee #2 for his/her time and effort in reviewing our manuscript, titled 'An improved Grassberger-Procaccia algorithm for analysis of climate system complexity' (ID: hess-2017-445). Your comments and suggestions are much appreciated. Please see our responses in the following section.

Comment 1. Section 2.1 Algorithm for Computing Correlation Dimension may be reduced as correlation dimension is relatively old.

Response: Thank you for this comment. Section 2.1 introduces the original G-P algorithm. We can point out the problems existing in the traditional algorithm. Furthermore, Section 2.2 is based on Section 2.1. To shorten Section 2.1, we will make following changes: (1) lines 77-78 will be removed ('The dimension of the time series of a variable is indicative of the number of factors governing the underlying dynamical processes'), and (2) lines 97-98 will be changed into: According to the relationship between $D_2(m)$ and $m$, the saturation value of $D_2(m)$ is defined as the correlation dimension.

Comment 2. Lines 117-119 and Figure 1: Authors compared equations in terms of y =0.5x. What is R square value for both the equations and this also can be taken into consideration while judging superiority of methods.

Response: Thank you for this suggestion. Indeed, adding R square value is better for evaluating the fitting results. We will add R square value in Figure 1 and lines 117-119.

Changes in manuscript:

It can be seen from Fig. 1 that the fitting line (y=0.60 x-0.068; $R^2$=0.854) obtained from the least squares method is seriously affected by outliers and deviated from the original line y=0.5 x. By contrast, the RANSAC method is able to distinguish the inliers from outliers effectively and results in a satisfactory fitting line (y=0.49 x+0.007; $R^2$=0.990), demonstrating the advantage of using the RANSAC algorithm for linear fitting.

[Figure]

**Figure 1: Comparison of the fitted lines obtained from the RANSAC algorithm and the least squares method.**

Comment 3. Line 125-140: Detailed information is provided and flow chart is also presented.

Detailed information can be reduced as the flow chart is self explanatory.

Response: Thank you for this comment. According to your suggestion, we will reduce the relevant section as follows.

Changes in manuscript:

The flow chart of the proposed procedures for calculating correlation dimensions is given in Fig. 2, which consists of five major steps. (1) For the time series $x(t)$, the time delay $\tau$ is computed by an autocorrelation function (Liebert and Schuster, 1989). Then set the minimum embedding dimension $m_{min}$=2 and reconstruct the phase space by increasing $m$ to obtain the correlation exponent function $C(r, m)$; (2) The normals of the scatter points on the ln $r$~ln $C(r, m)$ line are estimated via principal component analysis (Mitra et al., 2004); (3) The $K$-means clustering technique is performed on the normal set $N$ with $K$=2 to obtain two different clusters. Set a threshold value $T$ to determine the angle $\alpha$ between the two clusters. If $\alpha > T$, the data set with larger differences in normals is discarded. Then, the $K$-means clustering technique is repeated on the remaining data set until $\alpha \leq T$; (4) The RANSAC algorithm is used to fit a straight line through the set of remaining scatter points; and (5) The slope of the line obtained from the RANSAC method is computed to acquire the correlation dimension $D_2(m)$ for each $m$. Finally, the saturation correlation dimension is determined using the plot $D_2(m)$ vs. $m$.

Comment 4. Figure 8: More discussion will help to understand the figure effectively.

Response: Thank you for this comment. Considering that this is a technical paper, we limited our discussions for the purpose of brevity. We can give more discussion (i.e., the sentences in red color) about this figure.

Changes in manuscript:

[revised manuscript text omitted]

Wolf, A., Swift, J. B., Swinney, H. L.: Determining Lyapunov exponents from a time series. Physica D Nonlinear Phenomena, 1985, 16(3), 285-317.

---

## Author Comment (AC3) · 6 Sep 2018

We would like to thank Referee 1 for his/her time and effort in reviewing our manuscript, titled 'An improved Grassberger-Procaccia algorithm for analysis of climate system complexity' (ID: hess-2017-445). Your comments and suggestions are much appreciated. Please see our responses in the following section.

Comment 1. For readers to quickly catch your contribution, it would be better to highlight major difficulties and challenges, and your original achievements to overcome them, in a clearer way in abstract and introduction.

Response: Thank you for the comment here. In Introduction, we have stated some of the major problems associated with the current methods for computing correlation dimensions (e.g., in lines 11-12 'the use of this method is still not adaptive and relies heavily on subjective criteria', and in lines 53-54, 'However, the G-P method relies on visual inspections for choosing scaling regions, which is subject to human errors (Sprott and Rowlands, 2001)'. To deal with this important problem, we tried to find 'more objective and adaptive algorithms for identifying scaling regions to obtain more accurate estimates of correlation dimensions' (lines 59-60). Nonetheless, based on the reviewer's comment here, we will further highlight our contribution for computing correlation dimensions in Abstract and Introduction.

Changes in manuscript:

Lines 11-12 change into: However, the use of this method is still not adaptive and the choice of scaling regions relies heavily on subjective criteria.

Lines 57-61 change into: However, these existing methods for identifying scaling regions had the following problems: (1) the computing processes are still not adaptive and the choice of scaling regions relies heavily on subjective criteria, and (2) the use of the least squares method for fitting straight lines to determine correlation exponents can include outliers (Cantrell, 2008) and thus is not optimal.

Comment 2. It is shown in the reference list that the authors have several publications in this field. This raises some concerns regarding the potential overlap with their previous works. The authors should explicitly state the novel contribution of this work, the similarities and the differences of this work with their previous publications.

Response: Thank you for the comment. First, the novelty of our current work as compared to previous studies is discussed in details in Abstract and Methodology sections. Secondly, the studies cited in the reference list (presumably with the last name of Wang) are done by others and not published by the current authors.

Comment 3. It is mentioned in p.1 that an improved Grassberger-Procaccia algorithm is adopted for analysis of climate system complexity. What are the other feasible alternatives? What are the advantages of adopting this particular algorithm over others in this case? How will this affect the results? More details should be furnished.

Response: Thank you for the suggestion. In fact, we have given some alternative methods for studying climate system complexity, such as chaos theory, wavelet analysis, and dynamical analysis (line 35). In particular, for computing correlation dimensions, we also compared our newly proposed algorithm to two other commonly

used algorithms, namely the intuitive judgment and the point-based *K*-means clustering methods, based on two classical chaotic systems. Nevertheless, based on the reviewer's comment here, we will add few sentences to further illustrate the differences among existing methods.

Moreover, to address this comment as well as following comments made by the reviewer, we feel that in a single paper with limited space, it is not feasible and appropriate to include every aspect existing in the field of complexity analysis, which would deviate from the central theme of this study and make the manuscript unnecessarily excessive. In fact, there are several excellent books that are devoted to entirely discussing relevant problems, which we would like to refer the reviewer to (e.g., Bellie Sivakumar, 2017; Jayawardena, 2014). In addition, we will add those books to the reference list for the convenience of readers.

Sivakumar, B.: Chaos in Hydrology. Springer Netherlands, 2017.
Jayawardena, A. W.: Environmental and Hydrological Systems Modelling. Taylor and Francis Group, CRC Press, 2014.

Comment 4. It is mentioned in p.2 that Lorenz and Henon chaotic systems are adopted to test the effectiveness of the proposed algorithm for estimating correlation dimensions. What are the other feasible alternatives? What are the advantages of adopting these particular systems over others in this case? How will this affect the results? More details should be furnished.

Response: Thank you for this comment. Indeed, there are other chaotic systems (e.g., the Chen system (Chen and Ueta, 1999), and the Rössler system (Rössler, 1979)). Among those chaotic systems, the Lorenz and Henon with existing theoretical correlation dimensions have been mostly studied in the past, and thus used to analyze the chaotic behavior in climate systems and to test the effectiveness of algorithms for computing climate system complexity (e.g., Grassberger and Procaccia, 1983; Lai and Lerner, 1998; Ji et al., 2011). In our opinion, for the purpose of brevity and more importantly comparison among different studies and methods for computing climate system complexity, it is justified that standard systems, such as the Lorenz and Henon systems, should be adopted. Finally, the discussion on different chaotic systems is beyond the scope of this study. It would be unrealistic for us to compare all chaotic systems in one single paper. Certainly, we can add more details and the following references in our revised manuscript.

Changes in manuscript:

we will add the following sentences at the end of the line 146:

the Lorenz and Henon systems with existing theoretical correlation dimensions have been mostly studied in the past, and thus used to analyze the chaotic behavior in climate systems and to test the effectiveness of algorithms for computing climate system complexity (e.g., Grassberger and Procaccia, 1983; Lai and Lerner, 1998; Ji et al., 2011).

References:

Chen, G., Ueta, T.: Yet another chaotic attractor, International Journal of Bifurcation and Chaos, 9, 1465-1466, 1999.

Rössler, O. E.: An equation for hyperchaos. Physics Letters, 71A (2, 3): 155-157, 1979.

Lai Y. C., Lerner D.: Effective scaling regime for computing the correlation dimension from chaotic time series. Physica D, 115: 1-18, 1998.

Ji, C.C., Zhu, H. and Jiang, W.: A novel method to identify the scaling region for chaotic time series correlation dimension calculation, Chinese Sci. Bull., 56, 925-932, doi: 10.1007/s11434-010-4180-6, 2011.

Comment 5. It is mentioned in p.2 that the Haihe River Basin is adopted as the case study. What are other feasible alternatives? What are the advantages of adopting this particular case study over others in this case? How will this affect the results? The authors should provide more details on this.

Response: Thank you for the comment here. The reasons that we took the Haihe River Basin (HRB) as a case study are both practical and theoretical: (1) The HRB has been facing serious water shortage due to climate change and increasing water demands. Although previous studies have investigated the climate variability (e.g., rainfall, air temperature, and evaporation) in the HRB from different perspectives, to our best knowledge, there are still no attempts to quantify nonlinear characteristics of climatic variables, especially regarding their chaotic behaviors in the HRB, which is essential for understanding the nonlinearity of the climate system in the region; and (2) The HRB is a diverse hydroclimatic region with many sub-watersheds of varying geographical and hydroclimatic conditions, which make the region ideal for understanding the climate system complexity. Certainly, we will add more details about the advantages of adopting this particular case in our revised manuscript.

Changes in manuscript:

Add the following sentences in the line 218:

Although previous studies have investigated the climate variability (e.g., rainfall, air temperature, and evaporation) in the HRB from different perspectives, to our best knowledge, there are still no attempts to quantify nonlinear characteristics of climatic variables, especially regarding their chaotic behaviors in the HRB, which is essential for understanding the nonlinearity of the climate system in the region. Furthermore, the HRB is a diverse hydroclimatic region with many sub-watersheds of varying geographical and hydroclimatic conditions, which make the region ideal for understanding the climate system complexity.

Comment 6. It is mentioned in p.3 that the normal-based K-means clustering technique is adopted to partition all normals of the scatter points into K clusters with high similarity. What are other feasible alternatives? What are the advantages of adopting this particular technique over others in this case? How will this affect the results? The authors should provide more details on this.

Response: Thank you for the suggestion. We had provided some explanations in Section 2.2 and Section 3. The K-means clustering method is used to partition $n$

observations into *K* clusters. For each cluster, each observation belongs to the cluster with the nearest mean. In this paper, in order to find a precise scaling region, we used the normal based *K*-means clustering algorithm to remove the points that were obviously located outside of the real scaling region (See section 2.2). Different from previous *K*-means methods (e.g., the point-based *K*-means clustering method), we measured the similarity of points using the normal-based *K*-means clustering technique (e.g., quantifying the diversity between normals of different points). This is because the normal directions of different points in Figure 4(a) are greatly different. By comparison, the distance between points is much less, due to the use of the logarithmic scale that makes the points more densely distributed as ln *r* goes backward (see Fig 3(a)). Therefore, we proposed to use the normal-based *K*-means clustering algorithm. As a comparison, taking the classical chaotic models of Lorenz and Henon as two examples, the results obtained by our proposed normal-based *K*-means method outperformed those from the point-based *K*-means method (see Table 1). To illustrate this, we will add some sentences to show the advantages of normal-based *K*-means method.

Changes in manuscript:

We will add the following sentences in line 165:

Different from previous *K*-means methods (e.g., the point-based *K*-means clustering method), we measured the similarity of points using the diversity between normals of different points. The reason for using the normal-based method is that the directions of normals for different points may vary considerably (See Fig. 4b); whereas, for the point-based *K*-means method, the distance between different points might be small, making it difficult to separate the points into different clusters (Fig. 3a).

Comment 7. It is mentioned in p.4 that the Random Sample Consensus algorithm is adopted to fit a straight line through the log-transformed points. What are other feasible alternatives? What are the advantages of adopting this particular technique over others in this case? How will this affect the results? The authors should provide more details on this.

Response: Thank you for the comment. We have given the reasons for choosing the Random Sample Consensus algorithm (RANSAC) in section 2.2. As shown in section 2.2, the RANSAC algorithm outperformed the commonly used least squares method for linear fitting, based on a hypothetical example (Fig. 1).

Comment 8. It is mentioned in p.6 that the intuitive judgment method and the point-based K-means clustering method are adopted to compare the results obtained by the proposed method. What are the other feasible alternatives? What are the advantages of adopting these particular methods over others in this case? How will this affect the results? More details should be furnished.

Response: Thank you for the comment. The intuitive judgment method and the point-based *K*-means clustering method are two commonly used methods for identifying scaling region (e.g., Sprott and Rowlands, 2001; Ji et al., 2011). Although more comparisons can be done, additional comparisons may seem redundant. In

addition, it is unrealistic to list all the comparisons in one single paper.

Comment 9. It is mentioned in p.6 that the normal-based K-means clustering technique is adopted to determine the scaling regions of the curves in Fig. 3a. What are other feasible alternatives? What are the advantages of adopting this particular technique over others in this case? How will this affect the results? The authors should provide more details on this.

Response: This comment is the same as the comment 8.

Comment 10-11. 10. Some key parameters are not mentioned. The rationale on the choice of the particular set of parameters should be explained with more details. Have the authors experimented with other sets of values? What are the sensitivities of these parameters on the results? 11. Some assumptions are stated in various sections. Justifications should be provided on these assumptions. Evaluation on how they will affect the results should be made.

Response: Thank you for this comment. We rechecked the paper and found that the ranges of $r$ were missing. We will add more details in line 92. Other parameters have been given in the paper. We must point out that some of the parameters in this study were determined by routinely used methods. For example, the time delay (see line 86) was determined by the autocorrelation function. Some other parameters (for example, T=5 °) were determined by testing the data. In terms of the assumption about the value $r$, we will add it in our paper.

Changes in manuscript:
We will add more details in line 92:
Set $r_{min}$ and $r_{max}$ as the minimum and maximum distances between points, respectively (Ji et al, 2011; Lai and Lerner, 1998). If $r \leqslant r_{min}$, none of the vector points falls within the volume element and $C(r, m)=0$. Otherwise, if $r \geqslant r_{max}$, all vector points falls within the volume element and $C(r, m)=1$.

Comment 12. The discussion section in the present form is relatively weak and should be strengthened with more details and justifications.

Response: Considering that this is a technical paper, we limited our discussions for the purpose of brevity. We can give more details and justifications in our revised paper. Please see the following for details.

Changes in manuscript:
The spatial pattern of the correlation dimension for precipitation in the HRB may be largely attributed to the regional flow pathway of moisture fluxes, which is mainly controlled by the East Asian Summer Monsoon (EASM). The HRB is located in a monsoon-dominated region, where the EASM plays a leading role in the regional meteorological system. Chen et al. (2013) showed that EASM had significant impacts on the spatiotemporal distribution of precipitation in East China. Li et al. (2017) further suggested that there was a significant correlation between precipitation and the

EASM index in the HRB. Wang et al. (2011) revealed that large-scale atmospheric circulations had close relationships with precipitation patterns in the HRB by analyzing the moisture flux derived from NCAR/NCEP reanalysis data. Influenced by the large-scale atmospheric circulation, precipitation in the middle and southeast parts of the HRB is more sensitive to climate variability due to their locations closer to the ocean. This leads to the decreasing trend of precipitation from southeast to northwest in the HRB, suggesting that the supply of moisture for precipitation in the region mainly comes from the ocean.

Partly owing to the closer geographical proximity to the ocean (Fig. 8), the EASM has a stronger impact on precipitation in the southern and central areas than in the northern part of the HRB. Furthermore, at the north corner of the HRB, the Westerlies primarily affect the hydrometeorological system and thus weaken the impact of the EASM on precipitation (Li et al., 2017). In addition, other factors (e.g., topography, vegetation distribution, and human activity) may also have impacts on regional patterns of climate variables. In particular, the Yan-Taihang mountain located in the northwest HRB obstructs the vapor transport driven by the EASM, resulting in lower spatiotemporal variability in precipitation in the north part of the HRB. As a result, precipitation had higher degrees of complexity in the southern HRB, while its complexity was lower in the mountainous area in the northwest HRB. As to air temperature, the orographic effect in the mountainous area on air temperature might be stronger (Chu et al., 2010b), resulting in the higher complexity of temperature in this area. However, it should be noted that the range of the correlation dimension for air temperature from 1.0 to 2.0 suggests that two primary controls on temperature exist at all stations across the region.

Comment 13. The manuscript could be substantially improved by relying and citing more on recent literatures about real-life case studies of contemporary soft computing techniques in hydrological engineering such as the followings: Gholami, V., Chau, K. W., Fadaee, F., Torkaman, J., and Ghaffari, A. (2015). "Modeling of groundwater level fluctuations using dendrochronology in alluvial aquifers." J. Hydrol., 529, 1060–1069. Taormina, R., Chau, K.W., Sivakumar, B.: Neural network river forecasting through baseflow separation and binary-coded swarm optimization", Journal of Hydrology 529 (3): 1788-1797 2015. Wu, C. L., Chau, K. W., Fan, C.: Prediction of rainfall time series using modular artificial neural networks coupled with data-preprocessing techniques, Journal of Hydrology 389(1-2): 146-167, 2010. Wang W. C., Chau, K. W., Xu, D. M., Chen, X., Y., Improving forecasting accuracy of annual runoff time series using ARIMA based on EEMD decomposition, Water Resources Management 29 (8): 2655-2675 2015. Chen, X. Y., Chau, K. W., Busari, A. O., A comparative study of population-based optimization algorithms for downstream river flow forecasting by a hybrid neural network model," Engineering Applications of Artificial Intelligence 46 (A): 258-268 2015. Chau, K. W., Wu, C. L., "A Hybrid Model Coupled with Singular Spectrum Analysis for Daily Rainfall Prediction," Journal of Hydroinformatics 12 (4): 458-473 2010.

Response: Thank you for providing the relevant references for further modification of our paper, and we have read them and we will also cite some of them in the revised paper.

Comment 14. In the conclusion section, the limitations of this study, suggested improvements of this work and future directions should be highlighted.
Response: Thank you for this comment. We will add the limitations and future work of this study in the conclusion section.
Changes in manuscript:

The modified G-P algorithm proposed in this study can be used more objectively to describe the regionalization in the HRB, which has important significance in prediction in ungaged areas. Furthermore, the existence of chaotic behaviors of climate variables indicates that climate systems have deterministic types and are predictable in a short term. The accuracy of weather prediction can be improved by choosing reasonable number of influencing factors of climate system according to the correlation dimension values. It should be noted that more studies are still required to verify the present results using other nonlinear techniques, such as Lyapunov exponent (Wolf et al., 1985), and approximate entropy (Pincus, 1995). Besides, the improved G-P algorithm can be employed to analyze the nonlinear dynamics of other hydroclimatic variables, such as streamflow, soil moisture, and groundwater in the HRB and other regions. These results will be studied and reported in future.

References:
Cantrell, C. A.: Technical Note: Review of methods for linear least squares fitting of data and application to atmospheric chemistry problems, Atmos. Chem. Phys., 8, 5477-5487, 2008.
Li, F. X., Zhang, S. Y., Chen, D., He, L., and Gu, L. L.: Inter-decadal variability of the east Asian summer monsoon and its impact on hydrologic variables in the Haihe River Basin, China. J. Resour. Ecol., 8(2), 174-184, 2017.
Pincus, S.: Approximate entropy (ApEn) as a complexity measure. Chaos, 1995, 5(1), 110.
Wang, W. G., Shao, Q. X., Peng, S. Z., Zhang, Z. X., Xing, W. Q., An, G. Y., and Yong, B.: Spatial and temporal characteristics of changes in precipitation during 1957-2007 in the Haihe River basin, China. Stoch. Environ. Res. Risk Assess., 25(7), 881-895, 2011.
Wolf, A., Swift, J. B., Swinney, H. L.: Determining Lyapunov exponents from a time series. Physica D Nonlinear Phenomena, 1985, 16(3), 285-317.

---

## Author Response (AR1)

**Reply to Referee #1**

We would like to thank Referee #1 for his/her time and effort in reviewing our manuscript, titled 'An improved Grassberger-Procaccia algorithm for analysis of climate system complexity' (ID: hess-2017-445). Your comments and suggestions are much appreciated. Please see our responses in the following section.

**Comment 1**. For readers to quickly catch your contribution, it would be better to highlight major difficulties and challenges, and your original achievements to overcome them, in a clearer way in abstract and introduction.

**Response**: Thank you for the comment here. In Introduction, we have stated some of the major problems associated with the current methods for computing correlation dimensions (e.g., page 1, lines 11-12 of the manuscript 'the use of this method is still not adaptive and relies heavily on subjective criteria', and lines 53-54, 'However, the G-P method relies on visual inspections for choosing scaling regions, which is subject to human errors (Sprott and Rowlands, 2001)'. To deal with this important problem, we tried to find 'more objective and adaptive algorithms for identifying scaling regions to obtain more accurate estimates of correlation dimensions' (lines 59-60). Nonetheless, based on the reviewer's comment here, we highlighted our contribution for computing correlation dimensions in Abstract and Introduction.

**Changes in the manuscript**

Page 1, lines 11-12: However, the use of this method is still not adaptive and the choice of scaling regions relies heavily on subjective criteria.

Page 2, lines 57-60: However, these existing methods for identifying scaling regions had the following problems: (1) the computing processes are still not adaptive and the choice of scaling regions relies heavily on subjective criteria, and (2) the use of the least squares method for fitting straight lines to determine correlation exponents can include outliers (Cantrell, 2008) and thus is not optimal.

A reference was added on page 13, lines 329-331:

Cantrell, C. A.: Technical Note: Review of methods for linear least squares fitting of data and application to atmospheric chemistry problems, Atmos. Chem. Phys., 8, 5477-5487, https://doi.org/10.5194/acp-8-5477-2008, 2008.

**Comment 2**. It is shown in the reference list that the authors have several publications in this field. This raises some concerns regarding the potential overlap with their previous works. The authors should explicitly state the novel contribution of this work, the similarities and the differences of this work with their previous publications.

**Response**: Thank you for the comment. First, the novelty of our current work as compared to previous studies is discussed in details in Abstract and Methodology sections. Secondly, the studies cited in the reference list (presumably with the last name of Wang) are done by others and not published by the current authors.

**Comment 3**. It is mentioned in p.1 that an improved Grassberger-Procaccia algorithm is adopted for analysis of climate system complexity. What are the other feasible alternatives? What are the advantages of adopting this particular algorithm over others

in this case? How will this affect the results? More details should be furnished.

**Response**: Thank you for the suggestion. In fact, we have given some alternative methods for studying climate system complexity, such as chaos theory, wavelet analysis, and dynamical analysis (see page 1, lines 34-35). In particular, for computing correlation dimensions, we also compared our newly proposed algorithm to two other commonly used algorithms, namely the intuitive judgment and the point-based *K*-means clustering methods (see page 5, lines 154-155), based on two classical chaotic systems.

Moreover, to address this comment as well as following comments made by the reviewer, we feel that in a single paper with limited space, it is not feasible and appropriate to include every aspect existing in the field of complexity analysis, which would deviate from the central theme of this study and make the manuscript unnecessarily excessive. In fact, there are several excellent books that are devoted to entirely discussing relevant problems (e.g., Bellie Sivakumar, 2017). We added this book to the reference list for the convenience of readers.

**Changes in the manuscript**

Page 15, lines 415-416:

Sivakumar, B.: Chaos in Hydrology: Bridging determinism and stochasticity. Sydney, Springer: Netherlands, https://doi.org/10.1007/978-90-481-2552-4, 2017.

Page 4, 114-116 and page 6, lines 167-171: We added more details to describe the advantage of our improved method in our revised manuscript.

**Comment 4**. It is mentioned in p.2 that Lorenz and Henon chaotic systems are adopted to test the effectiveness of the proposed algorithm for estimating correlation dimensions. What are the other feasible alternatives? What are the advantages of adopting these particular systems over others in this case? How will this affect the results? More details should be furnished.

**Response**: Thank you for this comment. Indeed, there are other chaotic systems (e.g., the Chen system, and the Rössler system. Among those chaotic systems, the Lorenz and Henon systems with existing theoretical correlation dimensions have been mostly studied in the past, and thus used to analyze the chaotic behavior in climate systems and to test the effectiveness of algorithms for computing climate system complexity (e.g., Grassberger and Procaccia, 1983a; Lai and Lerner, 1998; Ji et al., 2011). In our opinion, for the purpose of brevity and more importantly comparison among different studies and methods for computing climate system complexity, it is justified that standard systems, such as the Lorenz and Henon systems, should be adopted. Finally, the discussion on different chaotic systems is beyond the scope of this study. It would be unrealistic for us to compare all chaotic systems in one single paper. According to your suggestion, we added more details and the following references in our revised manuscript.

**Changes in manuscript**

The following sentences were added on page 5, lines 145-148:

The Lorenz and Henon systems with existing theoretical correlation dimensions have been mostly studied in the past, and thus used to analyze the chaotic behavior in

climate systems and to test the effectiveness of algorithms for computing climate system complexity (e.g., Grassberger and Procaccia, 1983a; Lai and Lerner, 1998; Ji et al., 2011).

A reference was added on page 14, lines 369-270:

Lai, Y. C., Lerner, D.: Effective scaling regime for computing the correlation dimension from chaotic time series, Physica D, 115, 1-18, https://doi.org/10.1016/S0167-2789(97)00230-3, 1998.

**Comment 5**. It is mentioned in p.2 that the Haihe River Basin is adopted as the case study. What are other feasible alternatives? What are the advantages of adopting this particular case study over others in this case? How will this affect the results? The authors should provide more details on this.

**Response**: Thank you for the comment here. The reasons that we took the Haihe River Basin (HRB) as a case study are both practical and theoretical: (1) The HRB has been facing serious water shortage due to climate change and increasing water demands. Although previous studies have investigated the climate variability (e.g., rainfall, air temperature, and evaporation) in the HRB from different perspectives, to our best knowledge, there are still no attempts to quantify nonlinear characteristics of climatic variables, especially regarding their chaotic behaviors in the HRB, which is essential for understanding the nonlinearity of the climate system in the region; and (2) The HRB is a diverse hydroclimatic region with many sub-watersheds of varying geographical and hydroclimatic conditions, which make the region ideal for understanding the climate system complexity. We added more details about the advantages of adopting this particular case in our revised manuscript.

**Changes in manuscript**

Page 2, lines 68-75: Although previous studies have investigated the climate variability (e.g., rainfall, air temperature, and evaporation) in the HRB from different perspectives, to our best knowledge, there are still no attempts to quantify nonlinear characteristics of climatic variables, especially regarding their chaotic behaviors in the HRB, which is essential for understanding the nonlinearity of the climate system in the region. Furthermore, the HRB is a diverse hydroclimatic region with many sub-watersheds of varying geographical and hydroclimatic conditions, which make the region ideal for understanding the climate system complexity.

**Comment 6**. It is mentioned in p.3 that the normal-based K-means clustering technique is adopted to partition all normals of the scatter points into K clusters with high similarity. What are other feasible alternatives? What are the advantages of adopting this particular technique over others in this case? How will this affect the results? The authors should provide more details on this.

**Response**: Thank you for the suggestion. We had provided some explanations in Section 2.2 and Section 3. The $K$-means clustering method is used to partition $n$ observations into $K$ clusters. For each cluster, each observation belongs to the cluster with the nearest mean. In this paper, in order to find a precise scaling region, we used the normal based $K$-means clustering algorithm to remove the points that were

obviously located outside of the real scaling region (see Section 2.2). Different from previous *K*-means methods (e.g., the point-based *K*-means clustering method), we measured the similarity of points using the normal-based *K*-means clustering technique (e.g., quantifying the diversity between normals of different points). This is because the normal directions of different points in Figure 4(a) are greatly different. By comparison, the distance between points is much less, due to the use of the logarithmic scale that makes the points more densely distributed as ln *r* goes backward (see Fig 3(a)). Therefore, we proposed to use the normal-based *K*-means clustering algorithm. As a comparison, taking the classical chaotic models of Lorenz and Henon as two examples, the results obtained by our proposed normal-based *K*-means method outperformed those from the point-based *K*-means method (see Table 1). To illustrate this more clearly, we added some sentences to show the advantages of normal-based *K*-means method.

**Changes in manuscript**

The following sentences were added on page 6, lines 167-171: Different from previous *K*-means methods (e.g., the point-based *K*-means clustering method), we measured the similarity of points using the diversity between normals of different points. The reason for using the normal-based method is that the directions of normals for different points may vary considerably (See Fig. 4b); whereas, for the point-based *K*-means method, the distance between different points might be small, making it difficult to separate the points into different clusters (Fig. 3a).

**Comment 7**. It is mentioned in p.4 that the Random Sample Consensus algorithm is adopted to fit a straight line through the log-transformed points. What are other feasible alternatives? What are the advantages of adopting this particular technique over others in this case? How will this affect the results? The authors should provide more details on this.

**Response**: Thank you for the comment. We have given the reasons for choosing the Random Sample Consensus algorithm (RANSAC) in Section 2.2 (page 4, 114-116). As shown in Section 2.2, the RANSAC algorithm outperformed the commonly used least squares method for linear fitting, based on a hypothetical example (Fig. 1).

**Comment 8**. It is mentioned in p.6 that the intuitive judgment method and the point-based K-means clustering method are adopted to compare the results obtained by the proposed method. What are the other feasible alternatives? What are the advantages of adopting these particular methods over others in this case? How will this affect the results? More details should be furnished.

**Response**: Thank you for the comment. The intuitive judgment method and the point-based *K*-means clustering method are two commonly used methods for identifying scaling region (e.g., Sprott and Rowlands, 2001; Ji et al., 2011). Although more comparisons can be done, additional comparisons may seem redundant. In addition, it is unrealistic to list all the comparisons in one single paper.

**Comment 9**. It is mentioned in p.6 that the normal-based K-means clustering

technique is adopted to determine the scaling regions of the curves in Fig. 3a. What are other feasible alternatives? What are the advantages of adopting this particular technique over others in this case? How will this affect the results? The authors should provide more details on this.

**Response**: This comment is the same as the comment 6. We had provided some explanations in Section 2.2 and Section 3. To illustrate this more clearly, we added some sentences to show the advantages of normal-based $K$-means method.

**Changes in manuscript**

The following sentences were added on page 6, lines 167-171: Different from previous $K$-means methods (e.g., the point-based $K$-means clustering method), we measured the similarity of points using the diversity between normals of different points. The reason for using the normal-based method is that the directions of normals for different points may vary considerably (See Fig. 4b); whereas, for the point-based $K$-means method, the distance between different points might be small, making it difficult to separate the points into different clusters (Fig. 3a).

**Comment 10-11**. 10. Some key parameters are not mentioned. The rationale on the choice of the particular set of parameters should be explained with more details. Have the authors experimented with other sets of values? What are the sensitivities of these parameters on the results? 11. Some assumptions are stated in various sections. Justifications should be provided on these assumptions. Evaluation on how they will affect the results should be made.

**Response**: Thank you for this comment. We rechecked the paper and found that the ranges of $r$ were missing. We added more details in our revised manuscript (see lines 94-96). Other parameters have been given in the paper. We must point out that some of the parameters in this study were determined by routinely used methods. For example, the time delay (see line 86) was determined by the autocorrelation function. Some other parameters (for example, T=5°) were determined by testing the data. In terms of the assumption about the value $r$, we added it in our paper.

**Changes in manuscript**

The following sentences were added on page 3, lines 94-96: Set $r_{min}$ and $r_{max}$ as the minimum and maximum distances between points, respectively (Ji et al, 2011; Lai and Lerner, 1998). If $r \leqslant r_{min}$, none of the vector points falls within the volume element and C($r$, $m$)=0. Otherwise, if $r \geqslant r_{max}$, all vector points fall within the volume element and C($r$, $m$)=1.

**Comment 12**. The discussion section in the present form is relatively weak and should be strengthened with more details and justifications.

**Response**: Considering that this is a technical paper, we limited our discussions for the purpose of brevity. We added more details and justifications in our revised version. Please see the following for details.

**Changes in manuscript**

Add sentences on page 10, lines 261-262:

The HRB is located in a monsoon-dominated region, where the EASM plays a leading

role in the regional meteorological system.

Add sentences on page 10, lines 264-268:

Wang et al. (2011) revealed that large-scale atmospheric circulations had close relationships with precipitation patterns in the HRB by analyzing the moisture flux derived from NCAR/NCEP reanalysis data. Influenced by the large-scale atmospheric circulation, precipitation in the middle and southeast parts of the HRB is more sensitive to climate variability due to their locations closer to the ocean.

Add sentences on page 11, lines 272-279:

Furthermore, at the north corner of the HRB, the westerlies primarily affect the hydrometeorological system and thus weaken the impact of the EASM on precipitation (Li et al., 2017). In addition, other factors (e.g., topography, vegetation distribution, and human activity) may also have impacts on regional patterns of climate variables. In particular, the Yan-Taihang mountain located in the northwest HRB obstructs the vapor transport driven by the EASM, resulting in lower spatiotemporal variability in precipitation in the north part of the HRB. As a result, precipitation had higher degrees of complexity in the southern HRB, while its complexity was lower in the mountainous area in the northwest HRB.

Add the following references on page 14, lines 375-377, and page 15, lines 431-433:

Li, F. X., Zhang, S. Y., Chen, D., He, L., and Gu, L. L.: Inter-decadal variability of the east Asian summer monsoon and its impact on hydrologic variables in the Haihe River Basin, China, J. Resour. Ecol., 8(2), 174-184, https://doi.org/10.5814/j.issn.1674-764X.2017.02.008, 2017.

Wang, W. G., Shao, Q. X., Peng, S. Z., Zhang, Z. X., Xing, W. Q., An, G. Y., and Yong, B.: Spatial and temporal characteristics of changes in precipitation during 1957-2007 in the Haihe River basin, China. Stoch. Environ. Res. Risk Assess., 25(7), 881-895, https://doi.org/10.1007/s00477-011-0469-5, 2011.

**Comment 13**. The manuscript could be substantially improved by relying and citing more on recent literatures about real-life case studies of contemporary soft computing techniques in hydrological engineering such as the followings: Gholami, V., Chau, K. W., Fadaee, F., Torkaman, J., and Ghaffari, A. (2015). "Modeling of groundwater level fluctuations using dendrochronology in alluvial aquifers." J. Hydrol., 529, 1060–1069. Taormina, R., Chau, K.W., Sivakumar, B.: Neural network river forecasting through baseflow separation and binary-coded swarm optimization", Journal of Hydrology 529 (3): 1788-1797 2015. Wu, C. L., Chau, K. W., Fan, C.: Prediction of rainfall time series using modular artificial neural networks coupled with data-preprocessing techniques, Journal of Hydrology 389(1-2): 146-167, 2010. Wang W. C., Chau, K. W., Xu, D. M., Chen, X., Y., Improving forecasting accuracy of annual runoff time series using ARIMA based on EEMD decomposition, Water Resources Management 29 (8): 2655-2675 2015. Chen, X. Y., Chau, K. W., Busari, A. O., A comparative study of population-based optimization algorithms for downstream river flow forecasting by a hybrid neural network model," Engineering Applications of Artificial Intelligence 46 (A): 258-268 2015. Chau, K. W., Wu, C. L., "A Hybrid Model Coupled with Singular Spectrum Analysis for Daily Rainfall Prediction,"

Journal of Hydroinformatics 12 (4): 458-473 2010.

**Response**: Thank you for providing the relevant references for further modification of our paper, and we have read them and cited one of them in the revised paper.

**Changes in manuscript**

The following reference was added on page 15, lines 439-441:

Wu, C. L., Chau, K. W., and Fan, C.: Prediction of rainfall time series using modular artificial neural networks coupled with data-preprocessing techniques, J. Hydrol., 389(1-2), 146-167, https://doi.org/10.1016/j.jhydrol.2010.05.040, 2010.

**Comment 14**. In the conclusion section, the limitations of this study, suggested improvements of this work and future directions should be highlighted.

**Response**: Thank you for this comment. We added the limitations and future work of this study in the conclusion section.

**Changes in manuscript**

Add a paragraph on page 12, lines 301-310:

The modified G-P algorithm proposed in this study can be used more objectively to characterize the complexity of climate systems (and other hydrological systems, such as streamflow, soil moisture, and groundwater), and thus provide a more reliable estimate of the number of dominant factors governing climate systems. Theoretically, it can provide valuable information for optimizing the number of parameters in climate models to reduce computational demands and model parameter uncertainties. Furthermore, the findings of this study can be used for the regionalization of hydrometeorological systems in the HRB, which has important significance in prediction in ungaged areas (Lebecherel et al., 2016). It should be noted that more studies are still required to verify the present results using other nonlinear techniques, such as the Lyapunov exponent (Wolf et al., 1985) and the approximate entropy (Pincus, 1995), which might provide additional insights into climate complexity analysis.

Add the following references on page 14, lines 373-374, 400-401, and page 15, lines 437-438:

Lebecherel, L., Andreassian, V., and Charles, P.: On evaluating the robustness of spatialproximity-based regionalization methods, J. Hydrol., 539, 196-203, https://doi.org/10.1016/j.jhydrol.2016.05.031, 2016.

Pincus, S.: Approximate entropy (ApEn) as a complexity measure, Chaos, 1995, 5(1), https://doi.org/10.1063/1.166092, 110.

Wolf, A., Swift, J. B., Swinney, H. L.: Determining Lyapunov exponents from a time series, Physica D Nonlinear Phenomena, 1985, 16(3), https://doi.org/10.1016/0167-2789(85)90011-9, 285-317.

**Reply to Referee #2**

We would like to thank the Referee #2 for his/her time and effort in reviewing our manuscript, titled 'An improved Grassberger-Procaccia algorithm for analysis of climate system complexity' (ID: hess-2017-445). Your comments and suggestions are much appreciated. Please see our responses in the following section.

**Comment 1**. Section 2.1 Algorithm for Computing Correlation Dimension may be reduced as correlation dimension is relatively old.

**Response**: Thank you for this comment. Section 2.1 introduces the original G-P algorithm. We can point out the problems existing in the traditional algorithm. Furthermore, Section 2.2 is based on Section 2.1.

**Changes in manuscript**

To shorten Section 2.1, the following were revised:

(1) Remove the sentence: The dimension of the time series of a variable is indicative of the number of factors governing the underlying dynamical processes;

(2) Page 3, lines 101-102 are modified as: According to the relationship between $D_2(m)$ and $m$, the saturation value of $D_2(m)$ is defined as the correlation dimension.

**Comment 2**. Lines 117-119 and Figure 1: Authors compared equations in terms of y =0.5x. What is R square value for both the equations and this also can be taken into consideration while judging superiority of methods.

**Response**: Thank you for this suggestion. Indeed, adding R square value is better for evaluating the fitting results. We added R square value in Fig. 1 and in the text.

**Changes in manuscript**

Page 4, lines 121-123: line 121 (y=0.60 x-0.068; $R^2$=0.854) , line 123 (y=0.49 x+0.007; $R^2$=0.990)

[Figure]

**Figure 1: Comparison of the fitted lines obtained from the RANSAC algorithm and the least squares method.**

**Comment 3**. Line 125-140: Detailed information is provided and flow chart is also presented.

Detailed information can be reduced as the flow chart is self explanatory.

**Response**: Thank you for this comment. According to your suggestion, we reduced the relevant section.

**Changes in manuscript**

Page 5, lines 129-139 were revised as follows:

The flow chart of the proposed procedures for calculating correlation dimensions is given in Fig. 2, which consists of five major steps. (1) For the time series $x(t)$, the time delay $\tau$ is computed by an autocorrelation function (Liebert and Schuster, 1989). Then set the minimum embedding dimension $m_{min}=2$ and reconstruct the phase space by increasing $m$ to obtain the correlation exponent function $C(r, m)$; (2) The normals of the scatter points on the ln $r$~ln $C(r, m)$ line are estimated via principal component analysis (Mitra et al., 2004); (3) The $K$-means clustering technique is performed on the normal set $N$ with $K=2$ to obtain two different clusters. Set a threshold value $T$ to determine the angle $\alpha$ between the two clusters. If $\alpha>T$, the data set with larger differences in normals is discarded. Then, the $K$-means clustering technique is repeated on the remaining data set until $\alpha\leq T$; (4) The RANSAC algorithm is used to fit a straight line through the set of remaining scatter points; and (5) The slope of the line obtained from the RANSAC method is computed to acquire the correlation dimension $D_2(m)$ for each $m$. Finally, the saturation correlation dimension is determined using the plot $D_2(m)$ vs. $m$.

**Comment 4**. Figure 8: More discussion will help to understand the figure effectively.

**Response:** Thank you for this comment. Considering that this is a technical paper, we limited our discussions for the purpose of brevity. We added more discussion (i.e., the sentences in red color) about this figure.

**Changes in manuscript**

Add sentences on page 10, lines 261-262:

The HRB is located in a monsoon-dominated region, where the EASM plays a leading role in the regional meteorological system.

Add sentences on page 10, lines 264-268:

Wang et al. (2011) revealed that large-scale atmospheric circulations had close relationships with precipitation patterns in the HRB by analyzing the moisture flux derived from NCAR/NCEP reanalysis data. Influenced by the large-scale atmospheric circulation, precipitation in the middle and southeast parts of the HRB is more sensitive to climate variability due to their locations closer to the ocean.

Add sentences on page 11, lines 272-279:

Furthermore, at the north corner of the HRB, the westerlies primarily affect the hydrometeorological system and thus weaken the impact of the EASM on precipitation (Li et al., 2017). In addition, other factors (e.g., topography, vegetation distribution, and human activity) may also have impacts on regional patterns of climate variables. In particular, the Yan-Taihang mountain located in the northwest HRB obstructs the vapor transport driven by the EASM, resulting in lower spatiotemporal variability in precipitation in the north part of the HRB. As a result, precipitation had higher degrees of complexity in the southern HRB, while its

complexity was lower in the mountainous area in the northwest HRB.

Add the following references on page 14, lines 375-377, and page 15, lines 431-433:

Li, F. X., Zhang, S. Y., Chen, D., He, L., and Gu, L. L.: Inter-decadal variability of the east Asian summer monsoon and its impact on hydrologic variables in the Haihe River Basin, China, J. Resour. Ecol., 8(2), 174-184, https://doi.org/10.5814/j.issn.1674-764X.2017.02.008, 2017.

Wang, W. G., Shao, Q. X., Peng, S. Z., Zhang, Z. X., Xing, W. Q., An, G. Y., and Yong, B.: Spatial and temporal characteristics of changes in precipitation during 1957-2007 in the Haihe River basin, China. Stoch. Environ. Res. Risk Assess., 25(7), 881-895, https://doi.org/10.1007/s00477-011-0469-5, 2011.

**Comment 5**. Utility of estimation of correlation dimensions for the future work in HRB can be briefly mentioned.

**Response**: Thank you for this comment. We added the limitations and future work of this study in the conclusion section.

**Changes in manuscript**

Add a paragraph on page 12, lines 301-310:

[revised manuscript text omitted]